# Human Cripto-1 and Cripto-3 Protein Expression in Normal and Malignant Settings That Conflicts with Established Conventions

**DOI:** 10.3390/cancers16213577

**Published:** 2024-10-23

**Authors:** Frank Cuttitta, Josune García-Sanmartín, Yang Feng, Mary Elizabeth Sunday, Young S. Kim, Alfredo Martínez

**Affiliations:** 1Tumor Angiogenesis Unit, Mouse Cancer and Genetics Program, National Cancer Institute/Frederick Facility, Frederick, MD 21701, USA; fengya@mail.nih.gov; 2Angiogenesis Group, Oncology Area, Center for Biomedical Research of La Rioja (CIBIR), 26006 Logroño, Spain; jgarcias@riojasalud.es (J.G.-S.); amartinezr@riojasalud.es (A.M.); 3Duke University Medical Center, Durham, NC 27710, USA; mary.sunday@duke.edu; 4Cancer Prevention Science Branch, Division of Cancer Prevention, National Cancer Institute, Rockville, MD 20850, USA; kimyoung@mail.nih.gov

**Keywords:** Cripto-1, Cripto-3, pseudogene, oncofetal protein, monoclonal antibody, discrimination, cell signal transduction, carcinogenesis

## Abstract

A pseudogene, by definition, is a segment of DNA that structurally resembles an established gene but has undergone mutation drift or genomic translocation where it no longer codes for an active protein. An example of this type of relationship is seen with the oncofetal protein Cripto-1 (CR1) and its pseudogene homolog Cripto-3 (CR3). The CR1 gene is expressed on chromosome 3 and CR3 on the X chromosome. They both code for a 188AA protein but differ by six AAs. Given the closeness of their protein structure, commercially available antibodies cannot differentiate CR1 from CR3. Hence, CR3 remains a pseudogene in human biology and its relationship with tumor malignancies underdetermined. Our newly generated discriminatory anti-CR1/anti-CR3 monoclonal antibodies have now proven that CR3 is a translated protein found in diverse human cancers, tracking with disease severity, involved with cell signaling and expressed as a cell-tethered or soluble entity, thus negating its pseudogene status.

## 1. Introduction

In 1989, the cDNA for the human oncogene Cripto was initially isolated from an embryonic teratocarcinoma cell line (NTERA2), shown to transform NIH3T3 cells. This oncogene coded for a 188 amino acid (AA) protein (Cripto-1/CR1) and was located on chromosome 3 (3p21-p3) [1,2]. Two years later, the same group identified a pseudogene homolog to CR1, which maps to the X chromosome (Xq21-22) [3]. The cDNA for the pseudogene homolog lacked introns and coded for a predicted 188 AA protein (Cripto-3/CR3) that differs from CR1 by 6 AAs dispersed over the entire molecule [1,3]. Several additional Cripto pseudogenes have been found throughout the human genome, located in chromosomes 2q37, 6p25, and 19q13.1 [4]. However, only CR3 has been found to be translated into a biologically active protein in an artificial mouse F9 cell line model and to activate the Nodal signaling pathway [5]. Human CR1 has also been coined “Teratocarcinoma-Derived Growth Factor-1” (TDGF1) and human CR3 alternatively called “Teratocarcinoma-Derived Growth Factor-3” (TDGF3) or “Teratocarcinoma-Derived Growth Factor-1 pseudogene 3” (TDGF1P3) [6,7].

Over the past several decades, CR1 has been shown to be a multifunctional oncofetal protein regulating (1) implantation and fetal development [8,9,10,11,12], (2) maintenance of stem cell stability [13,14,15,16], (3) mammary gland development and lactation [17,18,19,20], (4) inflammation and immune response [21,22,23,24], (5) wound repair and tissue regeneration [25,26,27,28], (6) angiogenesis [29], and (7) carcinogenesis [30,31,32,33,34]. CR1 was shown to be highly expressed in human cancers of the lung, colon, breast, skin, stomach, pancreas, bladder, gall bladder, ovary, cervix, endometrium, prostate, and testis [30,32]. CR1 was proven to be a unique biological mediator that can function as a co-receptor in its membrane-bound form or as a receptor ligand in its enzymatically released soluble state [35,36]. CR1 is known to mediate tumor cell growth through a series of signal transduction pathways, including a Nodal-dependent Smad2/3 mechanism or Nodal-independent events encompassing Src, ras/raf/MAPK, and PI3K/Akt [21,30]. Distinct regions on the CR1 protein are known to interact with established binding proteins, mediating cell signaling events (Figure 1). These include the “EGF-like domain” (K_76_–R_111_) which complexes with Nodal and the “CFC” domain (C_115_–D_150_) that binds GRP78 or Alk4 [37]. It is to be noted that, of the multitude of publications cited above, none addresses or demonstrates differential CR1 versus CR3 protein expression in human biology.

If we take a quick historic overview of CR1 antibody production, we see a classic progress from polyclonal antibodies (PoAbs) > MoAbs > humanization of MoAbs > direct generation of humanized MoAbs via bio-panning technology. One of the first anti-CR1 reagents was reported in 1992, when Saeki et al. developed a rabbit PoAb targeting the EGF-Like domain of hCR1 using a peptide immunogen (CPPSFYGRNCEHDVRKE) conjugated to KLH to immunize the host animal [38]. IHC staining of human colorectal tumors was positive with this rabbit antibody, while a normal colon was reported to be negative [38,39]. Using the same immune reagent, human breast cancer cell lines and tumor tissue were stained positive for CR1, while adjacent non-involved tissue was negative [40,41]. Similar IHC staining with this rabbit antibody was reported for gastrin carcinoma and testis germ cell tumors [42,43]. A rat PoAb (C2-1) targeting the Gly_116_–Lys_127_ region of CR1 (CFC domain) was developed by Tsutsumi et al. and was shown to detect CR1 in urinary bladder and gall bladder/pancreatic tumors, in ductal cells of chronic pancreatitis tissue, and in some islet cells of the normal pancreas [44,45,46,47]. Two rat IgM MoAbs have been reported that bind to the EGF-like domain of CR1 and can detect this oncofetal protein in the cancer tissue of the colon/lung/breast but not in their respective normal counterparts [48]. These MoAbs suppressed human colon cancer cell line growth in vitro and in in vivo xenografts [48]. In breast cancer, IHC staining with these MoAbs has been shown to track with histological grade and patient survival [49]. Rabbit anti-CR1 polyclonal antisera have been used to identify molecular weight variation in this oncofetal protein based on the degree of glycosylation and to define the critical importance of fucosylation at position Thr_88_ within the EFG-like domain regulating Nodal signaling and bioactivity [50,51]. The Biogen company has dissected the entire human CR1 molecule with a series of mouse MoAbs, some of which targeted both the EGF-like domain and CFC domains and have proven to block human tumor cell growth in cell culture and in xenograft nude mouse models [52,53]. One of the Biogen mouse anti-CR1 MoAbs that binds to the N-terminus of both CR1 and CR3 was humanized and chemically linked to a potent tubulin inhibitor (the maytansinoid derivative, DM40) via a succinimide ester bridge [54]. This complex proved to be a stable payload circulating in the bloodstream, but once taken up by tumor cells, DM40 was released, disrupting tubulin and causing targeted cell death [54]. This reagent showed promising results in animal studies, and phase I/II trials were started but never completed (54). A recent landmark paper by Dr. Meg Duroux’s Denmark group compares commercially available anti-CR1 reagents (PoAbs and MoAbs) on their ability to recognize CR1 via IHC, IF, WB, and ELISA [55]. Attempting to target important conformational areas on CR1 critical for cell signaling and bioactivity, Focà et al. generated a mouse MoAb (1B4) that recognized a synthetic folded CFC (Lys112-Asp150) domain of the human oncofetal protein, having a 10^−10^ M K_D_ [56]. This antibody could induce a long-term in vitro inhibition of c-Src/MAPK/PI3K pathways in a human melanoma cell line [56]. Ishii and co-workers have introduced a new technological strategy to develop humanized anti-CR1 MoAbs via isolating clones having a high affinity to recombinant human CR1 thru bio-panning from a phage-display library [57]. Their clone 35 antibody had a 1.1 pmol affinity, could detect CR1 in both human tumor cell lines and tumor tissue, and could suppress the in vitro growth of human colon cancer cell line GEO [57]. In summary, an extensive cumulative study of anti-CR1 antibody development has been conducted over the past 30 years, yet none of the cited publications addressed the issue of differential CR1/CR3 protein expression, defining an obvious void in Cripto studies.

Thus, there is a critical need for said discriminatory reagents to unravel numbers of open questions, such as the following: Is CR3 translated into a bona fide protein in diverse human biological systems? Is CR3, like CR1, found as a tethered membrane-bound entity or enzymatically released as a soluble protein that could be detected in serum? Does CR3 bind and activate cell signaling pathways modulated by Nodal, GRP78, and Alk4 and compete with CR1 binding? What Cripto molecule is most prevalent in human cancers? A multitude of investigative questions remain unanswered due to the lack of selective antibodies. Hence, our focused effort to develop mouse anti-CR1 and anti-CR3 MoAbs to address these intellectual queries.

We present for the first time MoAbs that discriminate CR1/CR3, showing no measurable cross-reactivity by ELISA and having a K_D_ of 10^−8^ M to 10^−10^ M by Biacore determination. Our selective antibodies identify CR3 as a translated protein found in human tumor cell lines, in paraffin-embedded samples of human cancers and in serum from normal female donors and breast cancer patients. IHC staining results of pathological tissue reveal a diverse anatomical CR1/CR3 expression where some tumors express both proteins, others selectively express only CR3, and still others show that endothelial cells of the vascular tumor bed stain for CR1 while tumor cells express CR3. Lastly, we demonstrate that both CR1 and CR3 interact with established CR1-binding proteins Nodal, GRP78, and Alk4 and competitively interfere with one another for targeted binding. Our collective findings uncover a hereto unknown biological expression of CR3 as it relates to human tumorigenesis and opens new avenues of investigative science for this oncofetal protein.

## 2. Materials and Methods

### 2.1. Recombinant Proteins

R&D Systems (Minneapolis, MN, USA) reagents included human CR1/hCR1 (Cat# 145-CR/CF, Lot# HOG161806), murine CR1/mCR1 (Cat# 1538-CR/CF, Lot# IDQ0315011), human Cryptic/hCYP (Cat# 1410-CR/CF, Lot# HUU0108031), human NODAL/hNODAL (Cat# 3218-ND/CF, Lot# OLF1115091). GenScript Biotech (Piscataway, NJ, USA, hCR1 (Order No. U020FBG160-3/p4GE001) and hCR3 (Order No. U9958EJ280-2). MyBioSoure (San Diego, CA, USA) human CR3/hCR3 (Cat# MBS 1261262, Lot# 04238). Abcam (Cambridge, UK) human GRP78/hGRP78 (Cat# ab 78432, Lot# GR3208939-6), Creative BioMart, Ivanhoe North, Australia) human Alk4/hAlk4 (Cat# ACVR1B-64H, Lot# 166154).

### 2.2. Primary/Secondary Antibodies

Santa Cruz (Dallas, TX, USA) mouse anti-hCR1 MoAb (Clone H-10, Cat# sc-376448, Lot# I1113). R&D Systems mouse anti-hCR1 MoAb (Cat# MAB 2772, Lot# VABO 0117121). Cusabio rabbit anti-hCR3/TDGF1P3 PoAb (Cat# CBS-PA302689, Lot# 10114Y). Abcam (Cambridge, UK) reagents included rabbit anti-hCR1 MoAb (Cat# ab108391, Lot# GR96457-10), rabbit anti-hCR1 MoAb (Cat# 133236, Lot# GR96275-6), and mouse anti-GAPDH MoAb (Cat# ab8245/Lot# GR336823-1). GenScript Biotech (Piscataway, NJ, USA) mouse anti-hCR1 and anti-hCR3 MoAbs discussed below in M and M section. Jackson Laboratory (Bar Harbor, ME, USA) secondary detection reagents—donkey anti-mouse IgG-horseradish peroxidase (HRP) antibody (Cat# 711-034-150, Lot# 13222) and goat anti-rabbit IgG-HRP antibody (Cat# 111-035-144, Lot# 143556).

### 2.3. Pathological Human Tumor Tissue/Cancer Patient Serum Specimens

US Biomax (Derwood, MD, USA) was used as the source of all human microarray tissue analyzed by IHC staining. Lung CA (Cat# BC041115e), Breast CA (Cat# BC081116c), Colon CA (Cat# BC051110b), Prostate CA (Cat# BC19021a), and Ovarian CA (Cat# BC110118). All microarray tumor tissue came with corresponding sex, age, diagnosis, stage, Rx treatment regimen, and receptor data when appropriate for the anatomical tumor type. Normal donor and breast cancer patient serum samples were commercially available through ProteoGenex, Inc. (Inglewood, CA, USA). Tumor location, ethnicity, histological diagnosis, grade, stage, and TNM ranking was supplied by the vendor for each individual cancer patient serum sample. There were no corresponding “Personal Identifiable Information (PII)” revealed for these commercially available pathological tissues or sera, and they were given an NIH “Not Human Subject Research” rating. As such, these clinical samples were awarded an NIH IRB-Exempt waiver approval letter dated 12 January 2023.

### 2.4. Human Tumor Cell Lines and Immortalized Human Endothelial Cells

All human tumor cell lines were purchase through the American Type Culture Collection (ATCC, Manassas, VA, USA) and included MCF7/ATCC H-22 (human breast cancer, adeno CA), MDA-MB231/ATCC HTB-26 (human breast cancer, adeno CA, TNBC), HepG2/ATCC HB-8065 (human hepatocellular CA), HT29/ATCC HTB-38 (human colorectal adeno CA), and A549/ATCC CCL-186 (human bronchioloalveolar CA). hmVECs are immortalized human microvascular endothelial cells and were a gift from Dr. Rong Shao (University of Massachusetts, Boston, MA, USA).

### 2.5. Generating Selective Murine MoAbs Targeting Human CR1 vs. Human CR3

As a “Fee For Service”, we contracted GenScript Biotech (Piscataway, NJ, USA) to develop murine MoAbs that could discriminate human CR1 (Order No. U7733DJ240) from human CR3 (Order No. U6044DJ240) and vice versa. Using their proprietary computer program, GenScript targeted regions on the two molecules that were best suited as potential peptide immunogenic epitopes that would differentiate CR1 from CR3. As diagramed in Figure 1**,** these involved areas of proline (p) to leucine (l) shifts for CR1_65–75_ or aspartic acid (d) to tyrosine (y) shifts for CR3_40–50_. It should be noted here that similar AA mutational shifts in critical receptor structures have been associated with human disease and conformational alterations, thus implicating a potential area of interest in discriminating between similar bioactive homologs [58,59]. As shown in Figure 1, two peptides were generated for each target immunogen epitope chosen: (1) the CR1A peptide (qrvppmgiqhs) expressed on human Cripto-1 protein and the CR1B peptide (qrvlpmgiqhs) expressed on human Cripto-3 protein, and (2) the CR3A peptide (srgdlafrdds) expressed on human Cripto-3 protein and the CR3B peptide (srgylafrdds) expressed on human Cripto-1 peptide. Peptides CR1A and CR3A were conjugated to KLH, and BALB/c or C57BL6 mice were immunized. After several challenges, the resulting antibody titers were monitored using an ELISA strategy. Spleens were harvested, splenocytes isolated, Sp2/0 used as a fusion myeloma partner, and hybridomas successfully formed by electrofusion technology [60,61]. The initial clonal selection of the resulting antibody-producing hybridomas was accomplished by ELISA screening for high antibody production to the CR1A peptide and low or absent levels generated against CR1B. A similar hybridoma antibody-screening approach was used for the CR3A and CR3B peptides. A second-level screening ELISA strategy was used to make the final sorting determination of hybridomas making selective mouse anti-CR1 or anti-CR3 using solid-phase CR1A, CR1B, CR3A, and CR3B peptides and full-length recombinant human CR1 or CR3. Selective anti-CR1 MoAb-producing hybridomas were chosen based on their exclusivity to recognized CR1A and not CR1B peptides or to binding full-length CR1 but not CR3. Similarly, hybridomas making selective anti-CR3 MoAbs were chosen based on their ability to bind CR3A but not CR3B peptides or to recognize full-length CR3 and not CR1.

### 2.6. ELISA Assessment of Antibody Binding Characterization and Serial Titration Studies

All protein or peptide targets were passively adsorbed to 96-well polystyrene microtiter plates (Corning/Costar^®^3591, EIA/RIA flat bottom, medium binding, Thermo Fisher Scientific, Frederick, MD, USA, Cat# 07-200-36, Lot# 23816004). Target recombinant proteins were routinely solid-phased at 50 ng/50 μL/well diluted in Dulbecco’s PBS, incubated overnight at 4 °C, and blocked with Ham’s/F12 + 1%BSA + 0.1% Triton X100 solution (HFBTS). Primary antibodies were initially normalized to 0.1 mg/mL in the HFBTS diluent described previously, serially diluted from 1:100 to 1:204,800, added at (50 μL/well/dilution) to specified protein target plates, and incubated for 1 h on a rotary mixer at room temperature (RT). Detector secondary HRP antibodies (Jackson Laboratories) were diluted 1:5000 in HFBTS diluent, added 50 μL/well, and incubated on a rotary mixer for 30 min at RT. HRP substrate (TMB stable chromogen, Invitrogen Corporation, Carisbad, CA, USA, Cat# SB02/Lot#7390863A) was added 100 μL/well and incubated for 30 min at RT on a rotary mixer. The HRP/TMB enzymatic reaction was stopped by adding 50 μL/well 0.16N H_2_S0_4_ and the resulting blue > yellow color shift measured at 450 nm on a CLARIOstar reader (BMG LABTECH, Ortenberg, DE, USA). As with all analytical testing conducted in this manuscript, individual assays were performed in triplicate and representative results presented.

### 2.7. Biacore/Surface Plasmon Resonance

The binding of antibodies to CR1 or CR3 was assessed using a Biacore X100 instrument (Atlanta, GA, USA). CR1 or CR3 was diluted to 25 ng/mL in 10 mM sodium acetate buffer, pH 4.0, and immobilized on a CM5 sensor chip with an amine coupling kit. Approximately 350 RU CR1 or CR3 was immobilized on the detection chip. The reference flow cell was treated with the amine coupling reagent without exposure to the Cripto peptides. The running buffer was HBS-EP (10 mM HEPES, pH 7.4, 150 mM NaCl, 3 mM EDTA, 0.01% surfactant P20). The test antibodies were diluted in the running buffer to 0.8, 4, 20, 100, and 500 nM concentrations and assayed with the Single Cycle Program. The chip was regenerated with 10 mM acetic acid, pH 3.5, 1 M NaCl. The sensorgram was analyzed with BiaEvaluation software (Version 3.0), and data were fitted to a 1:2 binding model. 

### 2.8. Immunohistochemical (IHC) Staining of Normal and Malignant Human Tissue

After dewaxing, peroxidase activity was blocked with H_2_O_2_ in methanol, and sections were rehydrated through an ethanol series. Antigen retrieval was achieved with citrate buffer, pH6.0, 20 min at 95 °C. Normal donkey serum was added for 30 min to minimize background staining, and then the primary antibody was left overnight at a 1:750 dilution. The next day, after washing, a polymer was added that contains secondary antibodies plus peroxidases (Novolink Polymer, Novocastra Leica Biosystems). Then regular development with DAB was followed by a light counterstaining with hematoxylin.

### 2.9. Protein Extraction and Western Blot Analysis

Cell line samples were lysed in M-PER buffer (Thermo Fisher Scientific, Waltham, MA, USA) supplemented with a cOmplete ULTRA protease inhibitor cocktail (Roche Diagnostics, Basel, Switzerland). The protein content of the lysates was quantified using the Bradford Protein Assay (BioRad, Hercules, CA, USA). Nupage sample-reducing buffer (Invitrogen, Waltham, MA, USA) was added to 30 µg lysates, heated at 70 °C for 10 min, and separated by SDS polyacrylamide gel electrophoresis in 4–12% Bis-Tris gels (Invitrogen). Electrophoresed proteins were transferred to PVDF membranes using the iBlot 2 Transfer Device (Invitrogen). Once transferred, the membranes were blocked using 5% powder skim milk in Tris-buffered saline–Tween-20 (TBST; 25 mM Tris, pH 7.5, 150 mM NaCl, 0.1% (*v*/*v*) Tween-20). Membranes were exposed overnight to the primary antibodies (Anti-CR1/Clone 5G1-1/1:500dil, Anti-CR3/Clone 5G11-2/1:500dil, Santa Cruz Anti-Cripto/Cat# sc-376448/1:250 dil). On the next day, HRP-linked secondary antibodies (Jackson Laboratories, West Grove, PA, USA/Cat# 715-034-150/1:10,000) were applied for 1 h at RT. Peroxidase activity was detected with ECL Prime Western Blotting Detection Reagent (Cytiva, Marlborough, MA, USA) and captured in a ChemiDoc MP Imaging System (BioRad). The quantification of the immunoreactive bands was accomplished with ImageLab software, version 6.1 (BioRad). GAPDH was used as a loading control and detected with Abcam (Cambridge, UK) anti-GAPDH mouse MoAbs (Cat# ab8245/1:30,000dil).

### 2.10. CR1/CR3 Capture/Quantitative ELISA for Assessing Patient Sera

Abcam rabbit anti-CR1 MoAb (ab108391/Lot#GR96458-10) was passively adsorbed to a flat-bottom Costar 96-well polystyrene microtiter plate at 50 ng/50 μL/well diluted in PBS, incubated overnight at 4 °C, and blocked with HFBTS—this was used as the “Capture Plate”. Recombinant human CR1 (GenScript Biotech, Piscataway, NJ, USA) and CR3 (MyBioSource, San Diego, CA, USA) standard curves were made via serial dilutions starting with 50 ng/50 μL/well followed by 2-fold dilutions to 0.0244 ng/50 μL/well. Individual standard curve CR1 or CR3 concentration points were added to solid-phase Abcam antibody and run in triplicate for each point. Unknown serum samples were also run in triplicate on the same “Capture Plate”. Capture antibody-bound CR1 or CR3 proteins were detected with NCI anti-CR1 5G1-1 or NCI anti-CR3 5G11-2 MoAbs initially normalized to 0.1 mg/mL and then diluted to 1:3000 in HFBTS. Deposited mouse MoAbs were detected with Jackson Lab donkey anti-mouse IgG-HRP diluted 1:5000 in HFBTS. An enzyme immune complex was developed with TMB (Invitrogen), the colorimetric reaction was stopped with acid, and the resulting OD was read at 450 nm (CLARIstar, BMG LABTECH, Ortenberg, DE, USA). Based on the resulting values of our standard curve covering the OD450 measurements of our unknown serum samples, a straight-line formula was generated (y = mx + B where m = OD, X = slope, and B = “y” intercept value) and used to solve for the unknown serum value “y” as ng/mL.

### 2.11. ELISA Evaluation of CR1/CR3 Interaction to Established Binding Proteins (Nodal, GRP78, and Alk4)

As previously cited in Castro et al. [15], recombinant human binding proteins (Nodal, GRP78, and Alk4) were passively adsorbed to 96-welll microtiter plates at [50 ng/50 μL/well], incubated overnight at 4 °C, and blocked with HFBTS. Solid-phased Abcam rabbit anti-CR1 MoAbs ab108391 (N-terminal immune epitope) and ab133236 (C-terminal immune epitope) (Figure 1) were used as positive controls. GenScript hCR1 or MyBioSource hCR3 were added at [50 ng/50 μL/well] to the respective binding protein wells and incubated for 30 min at RT. Alternatively, hCR1 and hCR3 were mixed in equal concentrations [50 ng/50 μL/well], added to the respective solid-phase binding proteins, and incubated for 30 min at RT. Bound hCR1 or hCR3 was detected with NCI anti-hCR1 Clone 5G1-1 MoAbs or NCI anti-CR3 Clone 5G11-2 MoAbs initially adjusted to [0.1 mg/mL] and then diluted to 1:3000. Bound mouse primary MoAbs were detected with HRP antibody (Jackson Laboratories, donkey anti-mouse) diluted 1:5000 in HFBTS, which was added [50 μL/well] and incubated on a rotary mixer for 30 min at RT. HRP substrate (TMB stable chromogen, Invitrogen, Cat# SB02/Lot#7390863A) was added [100 μL/well] and incubated for 30 min at RT on a rotary mixer. The HRP/TMB enzymatic reaction was stopped by adding [50 μL/well] 0.16N H_2_S0_4_, and the resulting blue > yellow color shift was measured at 450 nm on a CLARIOstar reader.

### 2.12. Statistics

All individual tissue cores were evaluated for staining intensity (0–3) and staining distribution (0–3), 0 being no staining and 3 the maximum intensity or >90% positive cells. The sum of both parameters (0–6) determined the core’s value. The normality of the dataset distribution was assessed using the one-sample Kolmogorov–Smirnov test. Since the amount of data was small and the distribution was not normal, all datasets were compared with the Kruskal–Wallis one-way ANOVA test. A two-sided *p*-value lower than 0.05 was considered statistically significant. Analyses were performed with EZR software version R commander 1.68 (Saitama Medical Center, Jichi Medical University, Saitama, Japan), which is a graphical user interface for R (the R Foundation for Statatistical Computing, Vienna, Austria), and Prism, version 8.3.0 (GraphPad software, San Diego, CA, USA) for the preparation of graphs.

## 3. Results

### 3.1. Generation of Selective Mouse Anti-CR MoAbs That Differentiate CR1 and CR3

GenScript, as a “Fee for Service” contract (Price Quote 1530087/Project No. U7733DJ240), generated mouse MoAbs selective for hCR1 and demonstrating no cross-reactivity with hCR3, based on small sequence differences between proteins (Figure 1). Through a second GenScript contract (Price Quote 1530085/Project No. U6044DJ240) selective mouse MoAbs for hCR3 were generated having no measurable cross-reactivity with hCR1. Interestingly, although both BALB/C and C57BL/6N mouse stains were immunized identically with peptide/KLH conjugates, the C57BL/6N strain proved to be the better immune responder for both anti-CR1 and anti-CR3 antibody production. Initial screening strategies identified over 200 hybridoma clones for each targeted anti-CR1 vs. anti-CR3 MoAb. This number was sequentially reduced to 20 individual clones showing the highest selectivity for CR1 vs. CR3 (Table 1). After two rounds of subcloning, final sorting restrictions were used to isolate hybridoma clones of interest based on cell growth, antibody production, immunoglobulin isotype, and selective titers against commercially available recombinant CR1 and CR3. The following clones were identified as high-ranking prospects for selective binding to CR1: 5G1-1, 17G2-1, 10F7-1, 18A9-1, and 18E2-1. And the following for selective recognition of CR3: 1G10-2, 2D9-2, 2G1-2, 3B1-2, and 5G11-2 (Table 1). Of these two groups of hybridomas, one clone from each group was chosen based on competitive titration results from extensive characterization studies, and those were anti-CR1 NCI 5G1-1 and anti-CR3 NCI 5G11-2 (Figure 2A,B).

### 3.2. Determination of k_on_/k_off_/K_D_ Values by Biacore/Surface Plasmon Resonance for NCI Anti-CR1 and NCI Anti-CR3 MoAbs

Anti-CR1 NCI 5G1-1 and anti-CR3 NCI 5G11-2 hybridoma clones were expanded in DMEM/F12-10%FCS medium, and once at 50% density, the cells harvested, washed, and transferred to LONZA HL-1 hybridoma serum-free medium. Conditioned media (CM) were harvested after 48 h, filtered through a 0.45 mm vacuum filter, and mouse IgGs purified from the resulting supernatants via protein G columns (MillipreSigma HiTrap). To verify the specificity of each MoAb, surface plasmon resonance (SPR) analyses were performed. The resulting Biacore graphs generated for anti-CR1 NCI 5G1-1 MoAb, anti-CR3 NCI 5G11-2, and K_on_/K_off_/K_D_ values are presented (Figure 3A,B and Table 2); respective K_D_ values of 10^−8^ M and 10^−10^ M were observed. NCI 5G1-1 MoAbs only recognized immobilized CR1 and not CR3 at concentrations up to 500 nM. Similarly, NCI 5G11-2 MoAbs only recognized immobilized CR3 and not CR1, demonstrating a slow dissociation rate (k_off_), the hallmark of strong MoAb binding. Note that the Biacore graphs generated for NCI MoAbs comply with those values observed for our representative ELISA data (Figure 2 and Appendix A), thus validating consistency in our analysis.

### 3.3. Comparison Between Anti-CR1/Anti-CR3 NCI MoAbs and Commercially Available Anti-CR Products

As summarized (Table 3 and Appendix A), all five vendor anti-CR antibodies, which included those from Santa Cruz, R&D Systems, Cusabio, and Abcam and our PAN Rx MoAb (NCI 17G2-1), demonstrated cross-reactivity with both CR1 and CR3. Only the Abcam rabbit anti-CR1 MoAb (ab133236) targeting the EGF-like domain of CR1 cross-reacted with mouse CR1 (mCR1). Our MoAbs, NCI 5G1-1 and NCI 5G11-2, prove to be the only test reagents that selectively bound to their respective target antigens CR1/CR3, demonstrating no measurable cross-reactivity.

### 3.4. Immunohistochemistry (IHC) Results

Several human microarrays of normal/tumor tissue samples were purchased from US BioMax/Tissue Array. The vendor supplied the age, sex, pathological diagnosis, stage, grade, TMN-M/TMN-N data, and progesterone receptor levels when applicable without revealing any personal identifiable information (PII). The IHC analysis of human pathological tissue revealed a diverse anatomical staining pattern for CR1 and CR3 expression (Figure 4). The IHC staining of normal adult lung showed CR3 expression targeting the bronchial epithelium of the airway, which was devoid of CR1 staining (Figure 4(A1)). Several lung tumor tissue samples stained for both CR1 and CR3 (Figure 4(A2,A4)). Interestingly, in some lung tumors we found what we have defined as “Anatomical Separation”, where CR1 is expressed in the endothelial cells of the tumor vascular bed while CR3 is present in the main tumor body (Figure 4(A3,A5) and their respective enlargements, Figure 4(A3′,A5′)). Epithelial cells in the crypts of normal adult colon stained for both CR1 and CR3 (Figure 4(B1)). Some colorectal adenocarcinoma tumor cells stained for CR3 but not for CR1 (Figure 4(B3,B5)), and some stained for both CR1 and CR3 (Figure 4(B2,B4)). Interestingly, we again observed a variation of “Anatomical Separation” occurring, where CR1 was expressed in vascular endothelial cells but also in tumor cells along with CR3 (Figure 4(B2,B4) and their respective enlargements, Figure 4(B2′,B4′)). The ductal epithelium of the normal adult human breast stained for both CR1 and CR3 (Figure 4(C1)). Dual staining for CR1 and CR3 was also observed in multiple invasive ductal carcinomas (Figure 4(C2–C5)). Normal adult human ovary demonstrated a complex anatomical staining of CR1 and CR3 (Figure 4(D1–D3)). The mesothelium surrounding the ovary stained for both CR1 and CR3 (Figure 4(D1)). The pellucida in the primary follicles show intense targeted staining for CR1 and CR3, with the surrounding tissue having a muted staining (Figure 4(D2)). Cells of the inner theca stained for both CR1 and CR3 (Figure 4(D3)). These cells will develop into the lutein cells of the corpus luteus and produce androgens during pregnancy. Varied IHC-staining patterns were observed for ovarian tumors, where some cancers exclusively expressed CR3 (Figure 4(D4,D6)). In other tumors, both CR1 and CR3 were expressed, but CR3 always stained with higher intensity (Figure 4(D5,D7)). Normal prostate was negative for both CR1 and CR3 (Figure 4(E1)). Interestingly, normal-looking prostate tissue adjacent to a tumor stained for both CR1 and CR3, suggesting these proteins maybe an early marker for malignant transformation in precancerous cells (Figure 4(E2)). We observed several prostate tumors that were completely negative for both CR1 and CR3. However, in certain cancers we saw a dual expression of CR1 and CR3 (Figure 4(E4)), while in others we observed intense CR3 staining and muted or absent CR1 expression (Figure 4(E3,E5)). In 400+ cases of human tumor tissues examined by IHC, CR3 staining was always the most prevalent and most intense when compared to CR1. Furthermore, when tumor cells stained for both proteins, the staining was always cytoplasmic, and CR3 had a more granular pattern, whereas CR1 staining was more diffuse.

### 3.5. IHC Detection of CR1/CR3 in Human Tumor Tissue and Ranking of TMN-N/TMN-M/Grade/Stage/Progesterone Receptor Expression Based on Staining Intensity

Based on the pathological data supplied by US Biomax, we began to rank tumors based on their CR1 vs. CR3 staining intensity plotted against stage, grade, TMN_M/TMN_N values, and progesterone receptor levels where applicable. Prostate CA TMN_N0 vs. TMN_N1 ranking based on CR1 IHC staining demonstrated the ability to differentiate non-lymph node involvement vs. nearby lymph nodes having cancerous cells with a “*p*” value = 0.023 (Figure 5A). Evaluating CR1 expression in M0 vs. M1 prostate CA, here again we saw an ability to discriminate non-metastatic from metastatic cancers base on CR1 IHC staining intensity, with “*p*” value = 0.0094 (Figure 5B). Prostate CA staging as assessed by CR1 IHC staining intensity showed the higher the stage designation, the higher the observed CR1 expression (“*p*” = 0.0341) (Figure 5C). Hence, for prostate malignancies, it appears that disease severity parallels CR1 IHC tissue staining intensity. Colon CA grade levels +, ++, +++ compared to IHC CR3 staining intensity demonstrated an inverse to the grade—the lower the grade, the more intense the CR3 expression (“*p*” = 0.049 and 0.006, respectively) (Figure 5D,E). We revealed a progressive relationship between progesterone receptor (PR) expression levels on breast cancer tissue and the IHC staining intensity for CR3—the higher the PR levels, the higher the CR3 expression (“*p*” = 0.004) (Figure 5F). We present additional correlative relationships using IHC staining intensity to rank breast/colon tumors based on Ki67 grade and TMN_M/TMN_N/TMN_T values that proved to be not quite significant but trends in that direction (Appendix A–C). Also, an interesting “rise/fall” phenomenon was observed for CR3 expression in prostate CA tracking TMN_T values (T2/T3/T4) that is statistically significant, although we need to be cautious due to the low number of samples evaluated (Appendix A).

### 3.6. Western Blot Analysis of Human Tumor Cell Lines

Using our discriminatory anti-CR1 versus anti-CR3 MoAbs, we evaluated several human tumor cell lines for the expression of said oncofetal proteins. We present the resulting raw WB data for MCF7 (breast CA), MDA-MB231 (TNBC), HepG2 (hepatocellular CA), HT29 (colorectal CA), A549 (bronchioloalveolar CA), and hmVECs (immortalized human endothelial cells) (Figure 6A). Our anti-CR1 NCI 5G1-1 MoAb identified a band at 50 kDa MW that varied in intensity for all human tumor cell lines tested, although being almost absent in hmVEC cells. Conversely, our anti-CR3 NCI 5G11-2 MoAb detected a 32 kDa MW band that varied in intensity for all tumor cells evaluated (Figure 6A). For the combined WB data for CR1 versus CR3 and corresponding housekeeping GAPDH loading standard, see Figure 6A. Quantitative density measurements were assessed and normalized to GAPDH values as a loading control. The graphic representation of the normalized density measurements for human tumor cell lines in culture is shown (Figure 6B). It appears on average that CR1 represents the highest protein expression when compared to CR3 in most of the cell lines examined, with the exception of HepG2. Given the small sample size of human tumor cell lines examined by WB and even though CR1 appears to be more prevalently expressed than CR3, no definitive conclusions should be drawn until a more diverse and larger tumor cell line number is examined.

### 3.7. CR1/CR3 Capture/Quantitative ELISA Evaluation of Serum from Normal Female Donors Versus Serum from Breast Cancer Patients

The resulting standard curve for GenScript recombinant human CR1 captured on solid-phased rabbit anti-CR1 was detected with anti-CR1 NCI 5G1-1 (Figure 7A). The insert graph in Figure 7A represents the resulting straight-line and quantitation formula used to calculate CR1 values for unknown samples. The resulting standard curve for MyBioSource recombinant human CR3 captured on solid-phased rabbit anti-CR1 and detected with anti-CR3 NCI 5G11-2 is also shown (Figure 7B). The insert graph in Figure 7B represents the resulting straight-line and quantitation formula used to calculate CR3 values for unknown samples. The CR1 and CR3 levels observed for serum from normal female donors versus serum from breast cancer patients were represented with an n = 10 for both groups (Figure 7C,D respectively). Although there was no significant difference between the average CR1/CR3 values for serum samples from normal female donors versus breast cancer patients, possibly due to the small sample size, we do demonstrate the ability of our reagents to detect soluble CR1/CR3 in biological fluids.

### 3.8. CR1/CR3 Interaction with Solid-Phased Established CR1-Binding Proteins (Nodal, GRP78, and Alk4)

All targeted CR1-binding proteins were solid-phased, and CR1 or CR3 was introduced individually or mixed at equimolar concentrations. Solid-phase Abcam rabbit anti-CR1 MoAbs ab108391 and ab133236, both PAN Rx, were used as positive CR1/CR3-binding controls. CR1 bound to solid-phase binding proteins was detected with anti-CR1 NCI 5G1-1 MoAbs. Similarly, CR3 bound to solid-phase binding proteins was detected with anti-CR3 NCI 5G11-2 MoAbs. Adherent selective CR1/CR3 MoAbs were detected using donkey anti-mouse IgG-HPR. CR1 binds to solid-phase binding proteins and exogenous equimolar CR3 directly competed for CR1 binding **(**Figure 8A). CR3 binds to solid-phase binding proteins and exogenous equimolar CR1 directly competed for CR3 binding (Figure 8B). Graphs representing 100% bound CR1/CR3 and the resulting binding inhibition observed for the competing oncofetal proteins added are shown (Figure 8C,D). It should be noted that Abcam rabbit MoAb ab108391 is an extremely strong binder of CR1/CR3 (Table 3, Appendix A) and even though it has an immune epitope that recognizes common AAs of our anti-CR1/anti-CR3 MoAb target peptides (Figure 1), it did not interfere with the said binding, as noted in our capture/quantitative ELISA (Figure 5A,B) and CR1-binding protein assay (Figure 6A,B).

## 4. Discussion

We have developed mouse MoAbs that can differentiate CR1 from CR3 and have validated this ability using multiple characterization studies that include solid-phase ELISA testing, a Biacore (surface plasmon resonance) comparison, and IHC/WB analysis of Cripto negative/positive human cells. Multiple commercially available anti-human Cripto MoAbs all cross-react with CR1 and CR3 via the ELISA analysis. Our MoAb reagents proved to be unique in this testing series and showed no measurable cross-reactivity with CR1/CR3 from multiple sources, thus demonstrating their discriminatory capability. Neither our MoAbs nor the commercial products recognized solid-phase recombinant human Criptic, another member of the epidermal growth factor/Cripto family. Interestingly, of all the test reagents evaluated by ELISA, only the Abcam rabbit anti-human Cripto MoAb (ab133236) bound solid-phase mouse CR1.

Using NCI 5G11-2 MoAbs, we have confirmed that CR3 is a bona fide translated protein in human tissues, thus removing its prior pseudogene status. Our IHC studies for CR1/CR3 expression in paraffin-embedded samples of human tumor tissue have demonstrated diverse staining patterns where some tumors only have CR3, while other cases show a dual expression of both CR1 and CR3 in the tumor. Interestingly, we also define a new state of “Anatomical Separation” in a few human tumors studies that reveal CR1 expression in the endothelial cells of the tumor vascular bed and CR3 expression in the main tumor body. This was observed in certain human lung and colon CA samples. A similar anatomical phenomenology has been reported in glioblastoma tumors co-localizing CR1 (Millipore rabbit polyclonal anti-CR1 antibody) with CD31 (endothelial cell marker) via confocal microscopy, without mentioning issues of CR1 vs. CR3 expression [62]. What biological role could CR1 have in endothelial cells? It has been previously reported that CR1 functions as a tumor angiogenesis factor [29]. There are several examples in the literature that show that endothelial cells can produce a variety of peptide hormones that can directly regulate vascular growth and tone, for example endothelin-1 and adrenomedullin [63,64,65,66]. Hence it is not inconceivable that CR1 produced by endothelial cells could play a similar role in maintaining tumor vascular integrity.

We present IHC data in normal paraffin-embedded tissue that differ from what has been traditionally reported in the published literature. Most of the older studies reported negative IHC staining of normal tissue when using either rabbit or rat anti-CR1 PoAb or MoAb that targets either the EGF-like or CFC domains of this oncofetal protein [38,39,40,41,42,43,44,45,46,47,48,49]. Our selective anti-CR1/CR3 MoAbs bind to antigenic epitopes that are distal and N-terminal to older immune reagents’ recognition sites on CR1. In the normal adult state, CR1 levels are reported to be low and as such may be complexed with available established binding proteins such as Nodal (EGF-like domain) or GRP78 and Alk4 (CFC domain), hence interfering with older immune reagent binding (i.e., no IHC staining of normal tissue). As CR1 levels go up in inflammation or carcinogenesis, binding protein become saturated, and free CR1 molecules become available for older antibody interaction (i.e., IHC staining of tumor tissue). We and others have previously shown that antibodies that bind to either the EGF-like domain or CFC domain of CR1 block Nodal and Alk4 binding [21,52,56]. If said interaction were at an equilibrium, raising the concentration of either component would affect the binding of the other. Hence, describing what we feel is occurring in the normal setting, at low CR1/CR3 levels, EGF-like and CFC domains are saturated with established binding proteins, which inhibit older antibody interaction. That our selective anti-CR1/CR3 MoAbs can bind to their respective target immune epitopes on intact CR1 vs. CR3 proteins in the presence of Nodal/GRP78/Alk4 is confirmed by the resulting data in Figure 8.

CR1/CR3 both translate into a 188 AA protein. Considering an average molecular weight (MW) of 110 Da for an AA, we would expect an MW of about 21 kDa for native CR1/CR3. However, multiple higher MW forms of CR1 have been reported (36 kDa, 29 kDa, 27 kDa, and 24 kDa) due to glycosylation [50]. In fact, attached sugars play a critical role in CR1’s ability to activate the Nodal signaling pathway, and dyfucosylation of the EGF-like domain prevents CR1/Nodal interaction [51]. In our WB analysis of human tumor cell lines, a 50 kDa MW band for CR1 was consistently identified in all tested cells except for hmVEC extracts. A similar CR1 protein band of ≅50 kDa has been previously reported for human embryonic stem cell lysates using an Abcam (ab199170) rabbit anti-CR1 polyclonal antiserum [67]. Do the attached sugars play a role in this observed higher molecular weight form of CR1? This question could be addressed in future studies by suppressing glycosylation in vivo with either monensin or tunicamycin or directly cleaving sugar residues from CR1 products with N-glycosidase as was reported by Brandt et al. [50]. Speculating further, this high weight complex could possibly be a CR1-CR1 duplex formed by an alternative translational processing pathway that has yet to be identified. Finally, we have shown a 32 kDa band for CR3, indicating a different post-translational modification pattern for this protein, an event which deserves further investigation.

We report the development of a capture/quantitative ELISA for measuring CR1 and CR3 in biological fluids using Abcam rabbit anti-human cripto MoAbs (Ab108391) as the solid-phase capture entity. This MoAb binds CR1 and CR3 equally well via a common antigenic epitope (lafrddsiwpqeepairprssqr), (Figure 1 and Table 1). Using this quantitative ELISA, we can measure CR1 or CR3 in the picogram/mL range. The quantitative CR1 values we recorded for normal female donor serum (0.25 ± 0.119 ng/mL) were similar to reported levels in papers by Bianco et al. (0.32 ± 0.19 ng/mL), Pigaard et al. (0.59 ± 0.1 ng/mL), and Xu et al. (1.03 ± 0.36 ng/mL) [34,62,68]. On the other hand, our average quantitative data for breast cancer patients were not significantly different from our normal female controls and were considerably lower than those values published for human cancers such as breast CA (2.97 ± 1.48 ng/mL, n = 54), colon CA (4.68 ± 3.5 ng/mL, n = 33), and NSCLC (4.13 ± 1.38 ng/mL, n = 312) [34,62,68]. We feel the relatively low sample number for our evaluated breast CA sera may explain the discrepancy. In addition, there is a quantitative caveat with our assay design that should be mentioned. Given that the solid-phased capture MoAb used in our assay binds CR1 and CR3 equally well, the quantitative values determined will be relative, not absolute, due to the competitive nature of the assay. Higher CR1 levels will compete for CR3 binding, hence making the calculated CR3 levels lower than the actual, and a similar scenario would take place for higher CR3 levels. As a possible remedy for this built-in anomaly, we are now developing a quantitative ELISA that uses our selective MoAbs as the capture agent and the Abcam product as the detector, thus eliminating binding competition issues at the capture antibody site. Finally, CR3 quantitative values for normal female donor sera and for breast CA patient sera are again relative levels and subject to the same assay caveat mentioned above; however, these data do confirm that CR3 is a soluble entity that can be detected in the serum. Given the close AA similarity of CR1/CR3 and the identical nature of their common GPI-phospholipase D target site, one would expect the possibility of enzymatic cleavage and the release of tethered CR1/CR3, and our data validate this phenomenon.

Historically, CR1 is known to mediate cell signaling through Nodal/GRP78/Alk4-involved pathways [20,30,37]. Our analysis demonstrates that CR3 can also bind to these CR1-binding proteins and compete for CR1 interaction. Determining if CR3 mediates a biological response in human cell systems will be a critical issue for future investigations. Given that CR1/CR3 compete for binding protein recognition, it will be interesting to determine via Biacore analysis their respective K_D_ using Nodal/GRP78/Alk4-targeted chips.

## 5. Conclusions

The preponderance of published reports over the past thirty years target CR1 as the major oncofetal protein involved in embryogenesis, wound repair, and carcinogenesis, but our current data suggest that CR3 expression should also be further studied in these biological events. Historically, CR1 has been shown to be coded on chromosome 3 and its pseudogene homolog CR3, expressed on the X chromosome. These two oncofetal proteins are 188 AA in length and differ from one another by six AA dispersed out across the molecules. Given this close structural/antigenic similarity, the commercially available anti-CR1 reagents tested so far cross-react with CR1 and CR3. We are the first to report the development of mouse MoAbs that can discriminate between CR1 and CR3. The best of these reagents, the anti-CR1 NCI 5G1-1 MoAb and anti-CR3 NCI5G11-2 MoAb, were chosen based on their respective titration curves and Biacore affinity determinations. These reagents were then used for the IHC staining of pathological microarray cancer tissue, WB analysis of tumor cell lines, capture/quantitative ELISA analysis, and CR1/CR3 interactive binding protein studies. Our collective IHC data suggest that CR3 is more highly expressed in clinical specimens than CR1 and its staining is always more intense in human tumor tissue. Furthermore, the expression levels of CR1 and CR3 appear to track with disease severity. In addition, a newly coined terminology was established, “Anatomical Separation”, to describe a solid tumor phenomenon where CR1 stained endothelial cells of the vascular tumor bed while CR3 stained the main tumor body. Using our selective MoAbs for WB, we demonstrated that CR1 and CR3 are expressed by human tumor cell lines. Our capture/quantitative ELISA was able to detect CR1/CR3 at picograms/mL and demonstrated their expression in the sera from normal/cancer patient donors, establishing their soluble protein characteristics. Finally, our CR1/CR3-binding protein ELISAs reveal that these proteins could interact with solid-phased Nodal/GRP78/Alk4 and directly competed for binding. Our overall data present scientific proof that CR3 is a bona fide translated protein, expressed in normal and pathological tumor tissue and found as a soluble entity in sera, and it has potential biological activity given its ability to interact with established CR1-binding proteins.

## Figures and Tables

**Figure 1 cancers-16-03577-f001:**
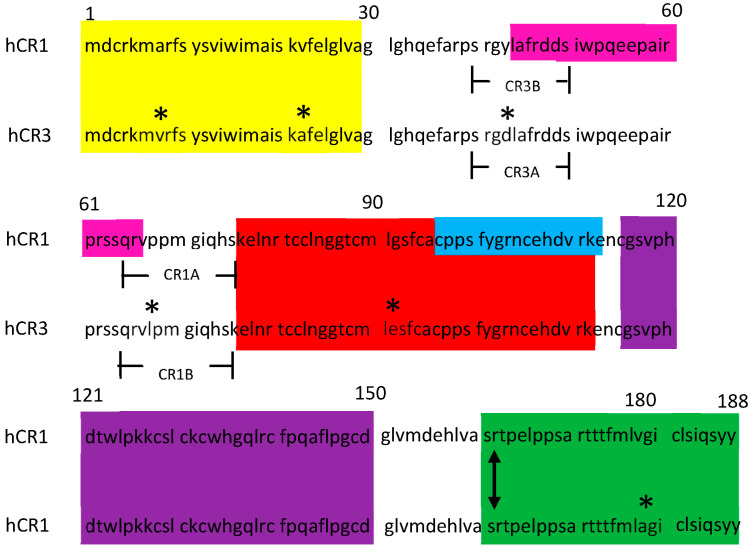
Amino acid (AA) sequence comparison of human CR1 (Accession No. AAH67844.1) and human CR3 (Accession No. AAG49539.1). The “*” indicates AA differences between CR1 and CR3. Yellow highlight = signal peptide. Fuchsia highlight = Abcam anti-CR1 MoAb ab108391 binding site (Leu44-Arg66). Red highlight = EFG-like domain (Nodal binding). Blue highlight = Abcam anti-CR1 MoAb ab133236 binding site (Cys97-Glu113). Purple highlight = CFC domain (GRP78/Alk4 binding). Green highlight = Glycoslylphosphatidylinositol (GPI) linkage domain. Double-headed arrow indicates the site of GPI-phospholipase D cleavage, resulting in the formation of soluble protein products. CR3A = peptide immunogen (srgdlafrdds) conjugated to KLH and used to generate anti-CR3 mouse MoAbs. CR3B = negative control peptide (srgylafrdds). CR1A = peptide immunogen (qrvppmgiqhs) conjugated to KLH and used to generate anti-CR1 mouse MoAbs. CR1B = negative control peptide (qrvlpmgiqhs).

**Figure 2 cancers-16-03577-f002:**
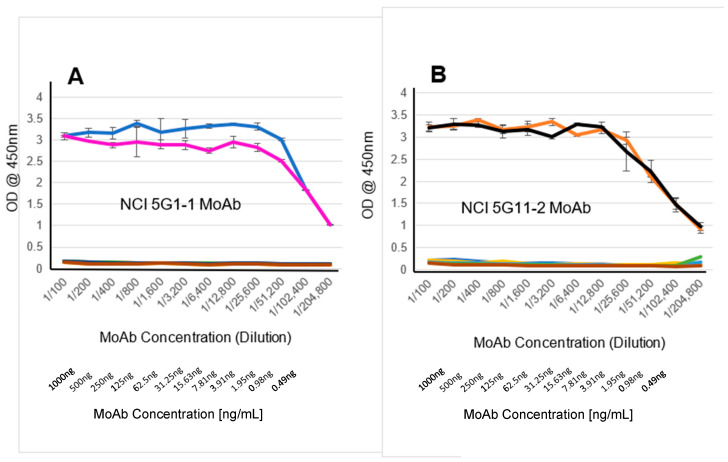
GenScript Biotech Corporation mouse MoAb production data and resulting titration curves for NCI final candidate choice. (**A**) Titration curve for anti-CR1 MoAb 5G1-1 evaluated on the following, solid-phased: Blue—R&D Systems recombinant human CR1, fuchsia—CR1A peptide immunogen, green—CR1B negative control peptide, gray—R&D Systems recombinant mouse CR1, yellow—R&D Systems recombinant human cryptic, orange—MyBioSource recombinant human CR3, black—CR3A peptide immunogen, and red—CR3B negative control peptide. (**B**) Titration curve for anti-CR3 5G11-2 evaluated on the following, solid-phased: Orange—MyBioSource recombinant human CR3, black—CR3A peptide immunogen, red—CR3B negative control peptide, blue—R&D Systems recombinant human CR1, fuchsia—CR1A peptide immunogen, green—CR1B negative control peptide, gray—R&D Systems recombinant mouse CR1, and yellow—R&D Systems recombinant human cryptic.

**Figure 3 cancers-16-03577-f003:**
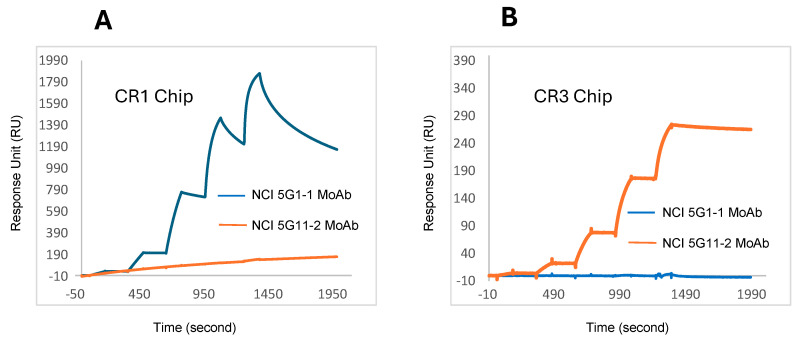
Biacore/surface plasmon resonance analysis of anti-CR1 NCI 5G1-1 MoAb and anti-CR3 NCI 5G11-2 MoAb binding to recombinant human CR1 or CR3. (**A**) NCI 5G1-1 MoAb (blue) diluted at 0.8, 4, 20, 100, and 500 nM versus NCI 5G11-2 (orange) at the same test concentrations measured interaction with immobilized CR1. (**B**) NCI 5G11-2 MoAb (orange) diluted at 0.8, 4, 20, 100, and 500 nM versus NCI 5G1-1 (blue) at the same test concentrations measured interaction with immobilized CR3.

**Figure 4 cancers-16-03577-f004:**
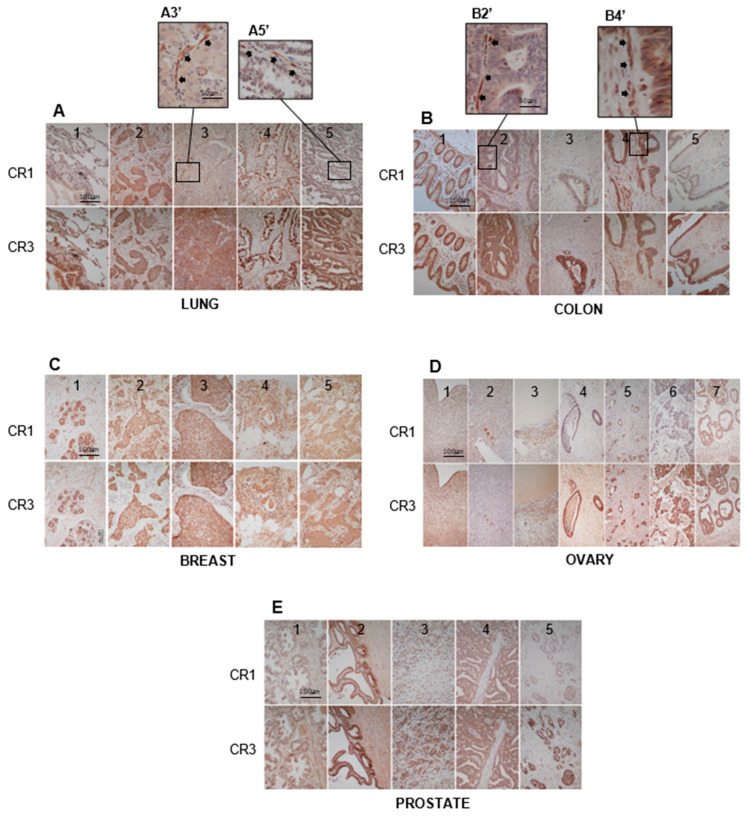
IHC staining of human normal/tumor tissue. (**A**) Lung IHC analysis: (**A1**) Normal lung, airway epithelium −CR3/+CR3, (**A2**) Squamous CA, +CR1/+CR3, (**A3**) Adenosquamous CA, tumor vascular endothelium +CR1/−CR3 (Anatomical Separation), tumor cells −CR1/+CR3, (**A3′**) Enlargement of +CR1 blood vessels—black arrowheads, (**A4**) Adenocarcinoma, +CR1/+CR3, (**A5**) Adenocarcinoma, tumor vascular endothelium +CR1/−CR3 (Anatomical Separation), and tumor cells −CR1/+CR3, and (**A5′**) Enlargement of +CR1 blood vessels—black arrowheads. (**B**) Colon IHC analysis: (**B1**) Normal colon, cryptic epithelium +CR1/+CR3, (**B2**) Adenocarcinoma, tumor vascular endothelium +CR1/−CR3 (Anatomical Separation Variant), tumor cells +CR1/+CR3, (**B2′**) enlargement of +CR1 blood vessels—black arrowheads, (**B3**) Adenocarcinoma, tumor cells −CR1/+CR3, (**B4**) Adenocarcinoma, tumor vascular endothelium +CR1/−CR3 (Anatomical Separation Variant), tumor cells +CR1/+CR3, (**B4′**) enlargement of +CR1 blood vessels—black arrowheads, and (**B5**) Adenocarcinoma, tumor cells −CR1/+CR3. (**C**) Breast IHC analysis: (**C1**) Normal breast, lobular ductal epithelium +CR1/+CR3, (**C2**–**C5**) Invasive ductal adenocarcinoma, tumor cells +CR1/+CR3. (**D**) Ovary IHC analysis: (**D1**) Normal ovary mesothelium +CR1/+CR3, (**D2**) Normal pellucida of the primary follicles +CR1/+CR3, (**D3**) Normal cells of the inner theca +CR1/+CR3, (**D4**) Endometrioid adenocarcinoma −CR1/+CR3, (**D5**) Metastatic adenocarcinoma +CR1/+CR3, (**D6**) Dysgerminoma −CR1/+CR3, and (**D7**) High-grade serous carcinoma +CR1/+CR3. (**E**) Prostate IHC analysis: (**E1**) Normal prostate −CR1/−CR3, (**E2**) Normal-looking tissue adjacent to adenocarcinoma +CR1/+CR3, possibly representing a premalignant lesion, (**E3**) Adenocarcinoma +/−CR1/+CR3, (**E4**) Adenocarcinoma +CR1/+CR3, and (**E5**) Adenocarcinoma −CR1/+CR3. Photographs taken at 20× magnification and size bar constant for all figures.

**Figure 5 cancers-16-03577-f005:**
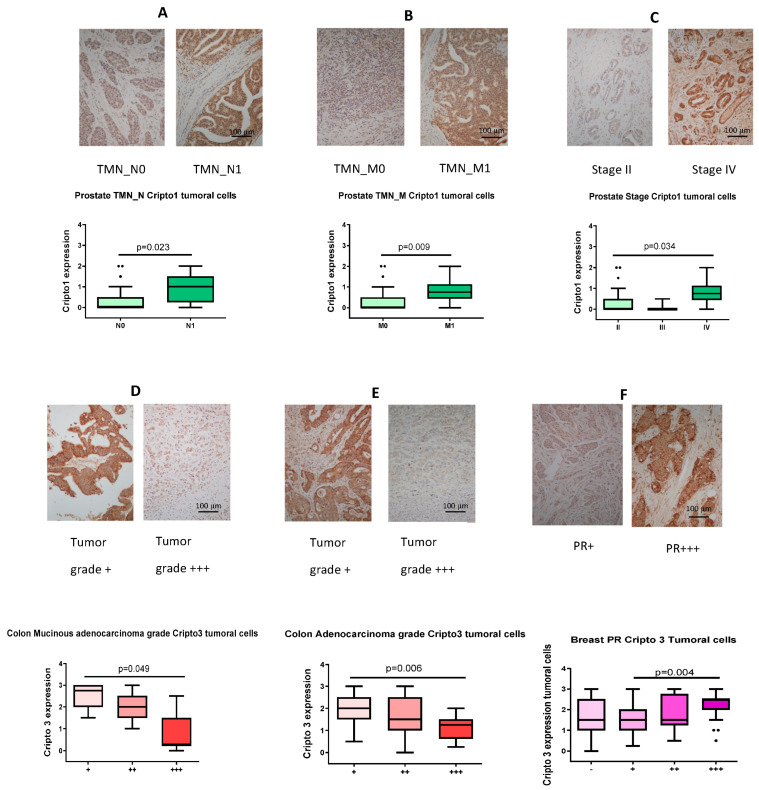
IHC detection of CR1/CR3 in human tissue and ranking of TMN_N, TMN_M, grade, stage, and progesterone receptor expression based on staining intensity. (**A**) Discriminate TMN_N0 from TMN_N1 in prostate cancer as scored by CR1 staining intensity. (**B**) Ranking of TMN_M0 vs. 5TMN_M1 in prostate cancer as revealed by CR1 staining. (**C**) Stage values for prostate cancer tracks with CR1 staining. (**D**) Discriminate colon mucinous adenocarcinoma grades +, ++, & +++ by CR3 staining. (**E**) Ranking of colon adenocarcinoma grades +, ++, and +++ based on CR3 staining. (**F**) Progesterone receptor score tracks with CR3 staining in breast cancer. Photographs taken at 20× magnification, and size bar constant for all figures. Dots indicate outliers.

**Figure 6 cancers-16-03577-f006:**
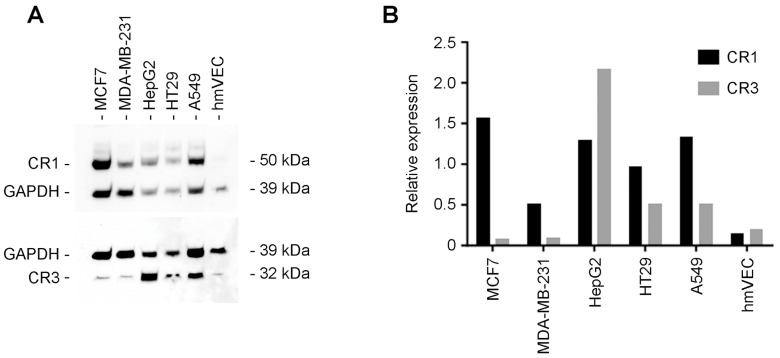
CR1 versus CR3 expression in human tumor cell lines as revealed by Western blot (WB) analysis. Note: CR1 and CR3 band intensity values normalized to respective GAPDH levels for the analyzed sample. Identification of human tumor cell lines and negative control. MCF7—breast CA, MDA-MB231—triple-negative breast CA, HepG2—hepatocellular CA, HT29—colorectal adeno CA, A549—bronchioloalveolar CA, hmVECs—immortalized endothelial cells. (**A**) Combined WB data for CR1 versus CR3 and corresponding housekeeping GAPDH loading standard. (**B**) Quantitative analysis of relative CR1 versus CR3 expression levels in cell lysates normalized to GAPDH loading control. Uncropped blots are shown in Appendix A.

**Figure 7 cancers-16-03577-f007:**
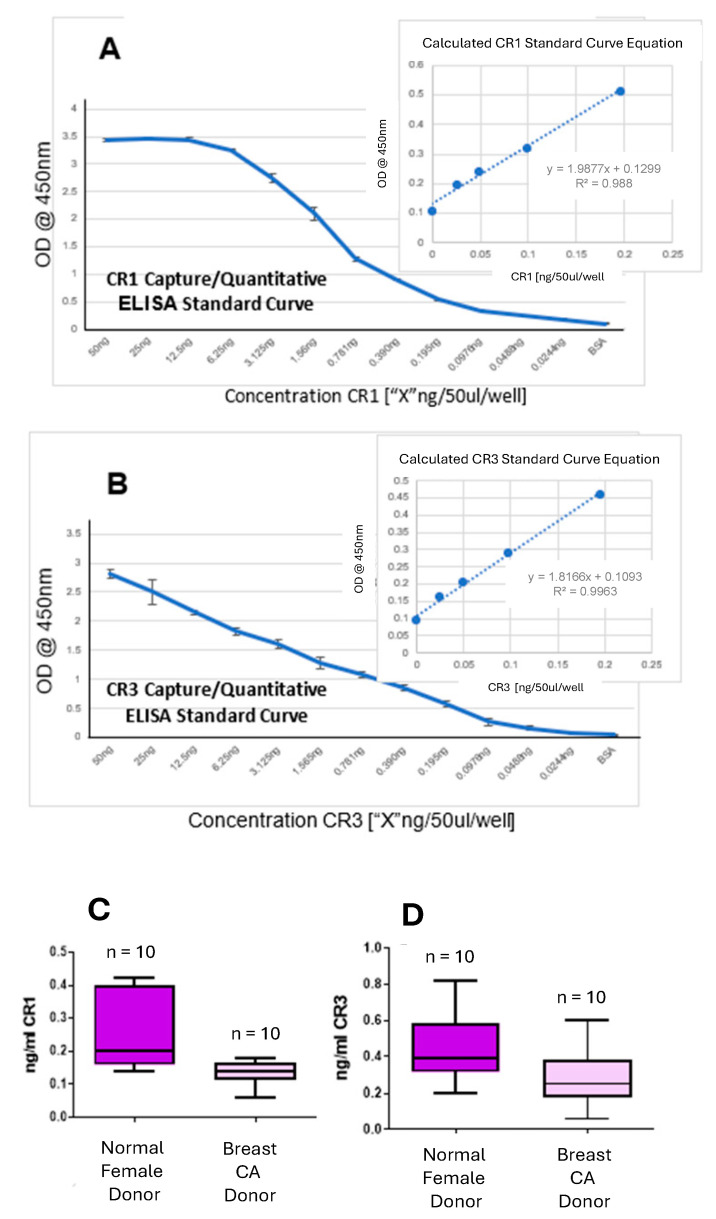
CR1/CR3 capture/quantitative ELISA for evaluation of soluble CR1/CR3 proteins in sera. (**A**) Representative CR1 capture/quantitative ELISA standard curve along with respective inserted linear equation graph used to determine CR1 values [ng/50 mL/well] of unknown serum sample. (**B**) Representative CR3 capture/quantitative ELISA standard curve along with respective inserted linear equation graph used to determine CR3 values [ng/50 mL/well] of unknown serum sample. (**C**) CR1 levels [ng/mL] identified in normal female donors and breast cancer patient sera. (**D**) CR3 levels [ng/mL] identified in normal female donors and breast cancer patient sera.

**Figure 8 cancers-16-03577-f008:**
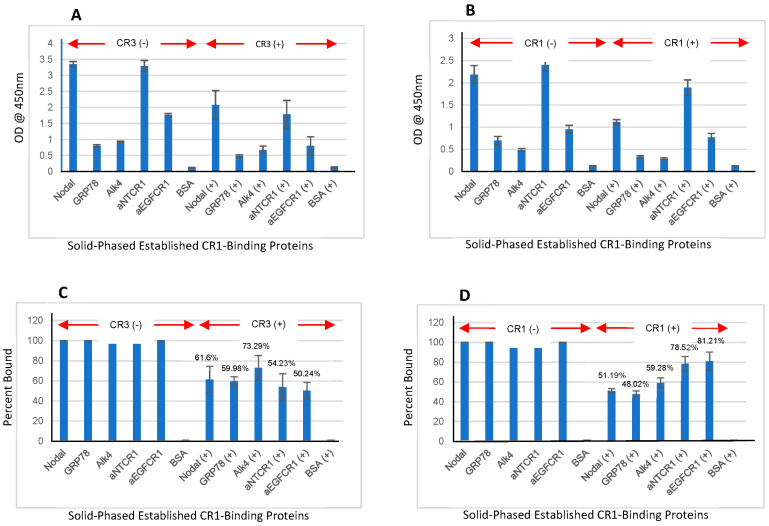
CR1/CR3 interaction with solid-phased established CR1-binding proteins (Nodal, GRP78, and Alk4). To validate our assay was working correctly, we utilized positive control solid-phased antibodies that recognized antigenic epitopes on CR1/CR3 distal from NCI MoAb binding sites. These included the Abcam anti-CR1 N-terminal target (ab108391) = aNTCR1 and the anti-CR1 EGF-like domain target (ab133236) = aEGFCR1. The interaction of CR1 or CR3 to solid-phased Nodal, GRP78, or Alk4 proteins was detected with, respectively, anti-CR1 NCI 5G1-1 or anti-CR3 NCI 5G11-2 MoAbs. (**A**) CR1 interaction with solid-phased binding proteins or controls in the presence or absence of equimolar CR3. (**B**) CR3 interaction with solid-phased binding proteins or controls in the presence or absence of equimolar CR1. (**C**) CR1 percent binding inhibition with equimolar CR3. (**D**) CR3 percent binding inhibition with equimolar CR1. All target proteins were solid-phased at [50 ng/50 μL/well] overnight and blocked with HFBTS.

**Table 1 cancers-16-03577-t001:** GenScript Biotech Corporation mouse anti-CR1 and anti-CR3 generation data.

		Screening	Positivity	Positivity in	Positivity		NCI Truncation	
	C57BL/6N	Scale 96-	in Primary	Confirmation	24-Well	Final	Best Hybridomas	
Order No.		Fusion No.	Wells/Plate	Screen	Screen	Plate	Delivery	5/20 Clones
U7733DJ240						Best 20	**5G1-1**,10F7-1	
(Anti-CR1)	#B1122	15 Plates	233 Clones	97 Clones	28 Clones	Clones	17G2-1, 18A9-1,	
							& 18E2-1	
U6044DJ240						Best 20	1G10-2, 2D9-2	
(Anti-CR3)	#B850	15 Plates	319 Clones	139 Clones	66 Clones	Clones	2G1-2, 3B1-2,	
							& 5G11-2	

Note: Highlighted clone in each anti-CR1 or anti-CR3 NCI truncation category represents the antibody chosen for extensive workup in this manuscript.

**Table 2 cancers-16-03577-t002:** Biacore/surface plasmon resonance analysis.

Monoclonal	Association Rate	Dissociation Rate	Dissociation Constant
Antibody Clone	K_on_ (1/Ms)	K_off_ (1/s)	K_D_ (M)
NCI 5G1-1	7.57 × 10^4^	2.45 × 10^−3^	3.23 × 10^−8^
NCI 5G11-2	5.33 × 10^4^	3.36 × 10^−5^	6.30 × 10^−10^

**Table 3 cancers-16-03577-t003:** Dilution of test antibodies giving 50% deflection of binding titration curve via solid-phase ELISA.

Anti-Cripto	R&D System	GenScript	GenScript	MyBioSource	R&D Systems	R&D Systems	BSA Neg	Binding
Reagents	Human CR1	Human CR1	Human CR3	Human CR3	Mouse CR1	Human Cryptic	Control	Selectivity
			Baseline					
NCI 5G1-1	1:51,200	1:51,200	Binding (BB)	BB	BB	BB	BB	CR1
	(1.9 ng/mL)	(1.9 ng/mL)						
NCI 511-2	BB	BB	1:51,200	1:51,200	BB	BB	BB	CR3
			(1.9 ng/mL)	(1.9 ng/mL)				
								CR1/CR3
NCI 17G2-1	1:25,000	1:12,800	1:12,800	1:12,800	BB	BB	BB	(Pan Rx)
	(3.9 ng/mL)	(7.8 ng/mL)	(7.8 ng/mL)	(7.8 ng/mL)				
Santa Cruz								
sc376448	1:3200	1:6400	1:6400	1:12,800	BB	BB	BB	Pan Rx
	(31.3 ng/mL)	(15.6 ng/mL)	(15.6 ng/mL)	(7.8 ng/mL)				
R&D Systems								
MAB2772	1:25,000	1:25,000	1:25,000	1:12,800	BB	BB	BB	Pan Rx
	(3.9 ng/mL)	(3.9 ng/mL)	(3.9 ng/mL)	(3.9 ng/mL)				
Abcam								
ab108391	1:102,400	1:51,200	1:102,400	1:204,800	BB	BB	BB	Pan Rx
	(0.98 ng/mL)	(1.9 ng/mL)	(0.98 ng/mL)	(0.48 ng/mL)				
Abcam								
ab133236	1:1600	1:12,800	1:25,000	1:51,200	1:3200	BB	BB	Pan Rx
	(62.5 ng/mL)	(7.8 ng/mL)	(3.9 ng/mL)	(1.95 ng/mL)	(31.2 ng/mL)			
CUSABIO								
PA302689	1:800	1:3200	1:6400	1:12,800	BB	BB	BB	Pan Rx
	(125 ng/mL)	(31.2 ng/mL)	(15.6 ng/mL)	(7.8 ng/mL)				

As indicated in the top row, recombinant Cripto target proteins were solid-phased at [50 ng/50 μL/well] and first-column test antibodies adjusted to [0.1 mg/mL] before being serially diluted (1:100–1:204,800 or 1000 ng/mL–0.49 ng/mL). Reference Figure 2 for direct conversion of dilution to ng/mL values. See Appendix A for complete titration curves.

## Data Availability

The data presented in this study are available in this article and Appendix A.

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
