# Peer review of "Human Cripto-1 and Cripto-3 Protein Expression in Normal and Malignant Settings That Conflicts with Established Conventions"

_cancers, 2024, doi:10.3390/cancers16213577_

Round 1

Reviewer 1 Report

Comments and Suggestions for Authors

This manuscript by Cuttitta et al. describes the generation and characterization of monoclonal antibodies that discriminate between two highly homologous proteins, CR1 and CR3. These proteins may play an important role in cancer development, and no previously existing antibodies were able of discriminate between them, to the point that CR3 encoding sequence in the genome is considered a pseudogene.

Furthermore, by using these new antibodies in different applications (IHC, ELISA), they confirm that CR3 is a product of the translation of the corresponding gene and, additionally, can show a differential expression pattern with respect to CR1. This finding is a novelty and may have profound implications in the field of the functions of CRIPTO proteins in tumour development.

The characterization data of the new antibodies are robust and show their ability to distinguish between CR1 and CR3, so the IHC and ELISA techniques in which they are used can be considered reliable. Therefore, the results obtained would be indicative of the possible importance of the CR3 protein in tumor development, although they should be taken as preliminary.

I believe that the manuscript describes tools and data that may be of great interest to potential readers and deserves to be published. However, I would like to make some observations to the authors in order to improve and clarify some points.

Minor points:

·      The quality of the figures needs to be improved. There is pixelation, especially in the text.

·         In lines 225, 279, and 299, "absorbed" is used. The correct term to use is "adsorbed".

·         Table 1 is split between pages 8 and 9, making it difficult to understand. Additionally, I think the column headers could be clearer in terms of formatting.

·         The paragraph starting on line 329 is split by Table 1. Line 342 stops suddenly and starts again within the footer of Table 1.

·         Table 2 and Table 3 could be formatted so that the header aligns with the columns.

·         The paragraph starting on line 414 is split by Table 3. Line 416 stops suddenly and starts again on line 453.

·         Lines 493 to 510 in the main text should be part of the Figure 4 footer.

·         Figure 5 is large and splits between pages 15 and 16. This could be a barrier to interpret the  information it contains. Could it be possible to rearrange the images and charts so that they all fit on one page?

·         Figure 6A appears the same as the original supplemental WB image gel, which should be provided without marks and uncut.

·         Results section 3.7 is entitled CR1/CR3 Capture/Quantitative ELISA Evaluation of Serum from Normal Female Donors Versus Serum from Breast Cancer Patients or Conditioned Media from Human Tumor Cell Lines, but no data on cell culture conditioned media are shown.

·         The paragraph starting on line 606 is split by Figure 8. Line 620 stops suddenly and starts again on line 634.

Additional comments:

·         I am surprised by the large size of the CR1 protein in WB. The authors point out in the Discussion section that glycosylation may explain this increase in molecular weight compared to the theoretical one for a 188 aa peptide. Although the electrophoretic separation was performed under reducing conditions, which should rule out disulfide bonds, could some kind of dimerization be possible?

·         Regarding the capture ELISA procedure for measuring CR1 and CR3 in biological fluids, the authors argue (lines 710-714) that the competitive nature of the assay due to the use of a capture antibody that binds equally to CR1 and CR3 in a nonspecific manner, produces relative results that may be biased if there is a large amount of one of the proteins in the sample. To address this issue, it is stated (lines 715-717) that quantitative ELISA assays using the new CR1- and CR3-specific monoclonal antibodies as capture reagents are being developed. Do the authors have preliminary data on the performance of these assays and the magnitude of potential bias with the original ELISA setup?

Comments on the Quality of English Language

The use of English in the manuscript is fine. It only requires a revision to fix typos.

Author Response

Reviewer 1

Reviewer's Concern 1 - Change the use of "absorbed" in lines 226/279/299 to the corrected grammar of "adsorbed"

Author's Correction - All suggested grammatical mistakes have been corrected

Reviewer's Concern 2 - Table 1 is split between pages 8 and 9, making it difficult to understand.  Additionally, I think the column headers could be clearer in terms of formatting.

Author's Correction - Formatting of Table 1 has been corrected and now displayed on a single page. 

Reviewer's Concern 3 - The paragraph starting on line 329 is split by Table 1.  Line 342 stops suddenly and starts again within the footer of Table 1.

Author's Correction -Appropriate corrections have been completed to satisfy Reviewer's 1concerns.

Reviewer's Concern 4 - Table 2 and Table 3 could be formatted so that the headers aligns with the columns.

Author's Correction - Table 2 and Table 3 have been reformatted to reflection Reviewer's 1 concerns.

Reviewer's Concern 5 - The paragraph starting on line 414 is split by Table 3.  Line 416 stops suddenly and starts again on line 435.

Author's Correction - Appropriate corrections have been made to address Reviewer's 1 concerns. 

Reviewer's Concern 6 - Line 493 to 510 in the main text should be part of the Figure 4 footer. 

Author's Correction - Appropriate corrections have been made to address Reviewer's 1 concerns. 

Reviewer's Concern 7- Figure 5 is large and splits between pages 15 and 16.  This could be a barrier to interpret the information it contains.  Could it be possible to rearrange the images and charts so that they fit on one page?

Author's Correction - Figure 5 has been truncated to accommodate a smother transition between pages. 

Reviewer's Concern 8- Figure 6A appears the same as the original supplemental WB image gel, which should be provided with without marks and uncuts.

Author's Correction - They are the same gel provided by Dr. Alfredo Martínez's co-author's group.

Reviewer's Concern 9- Results section 3.7 is entitled CR1/CR3 Capture/Quantitative ELISA Evaluation of Serum for Normal Female Donors Versus Serum from Breast Cancer Patients and Conditioned Media from Human Tumor Cell Lines, but not data on cell culture conditions media are shown.

Author's Correction - The reference to "Conditioned Media from Human Tumor Cell Lines" has been removed from Results section 3.7 title heading.  This data will be part of a future manuscript describing an alternative quantitative ELISA we are developing that uses solid phased anti-CR1 NCI 5G1 MoAb and anti-CR3 NCI 5G11-2 as the capture antibody and Abcam rabbit anti-CR1 MoAb (ab108391) as the detector component. 

Reviewer's Concern 10- The paragraph staring on line 606 is split by Figure 8.  Line 620 stops suddenly and starts again on line 634.

Author's Correction - Figure 8 has been truncated and no longer slips main text. 

Reviewer's Concern 11-I am surprised by the large size of the CR1 protein in WB.  The authors point out in the Discussion section that glycosylation may explain this increase in molecular weight compared to the theoretical one of a 188 aa peptide.  Although the electrophoresis separation was performed under reducing conditions, which should rule out disulfide bonds, could some kind of dimerization be possible. 

Author's Correction - Reviewer 1 suggest that an unknown CR1-CR1 dimerization processing event could be takin place which in not S-S dependent could explain the higher 50kDa WB product.  In our modified "Discussion" section we cite this potential possibility and also suggest in future WB analysis of increased numbers and more divers human tumor cell lines to use monensin/tunicamycin to block in vivo glycosylation or N-glycosidase to directly cleave sugar residues as was reported by Brandt et al. (reference 50 of the manuscript). 

Reviewer's Concern 12-Regarding the capture ELISA procedure for measuring CR1 and CR3 in biological fluids, the author argue that the competitive nature of the current assay due to the use of a capture antibody that binds to CR1 and CR3 in a nonspecific manner, produces relative results that may be biased if there is a larger amount of one of the proteins in the sample.  To address this issue, it is stated that quantitative ELISA assays using the new CR1- and CR3- specific monoclonal antibodies as capture reagents are being developed.  Do the authors have preliminary data on the performance of these assays and the magnitude of potential bias with the original ELISA setup?

Author's Correction - As stated in response to "Reviewer's Concern 9", this data will be part of a future manuscript describing an alternative quantitative ELISA we are developing that uses solid phased anti-CR1 NCI 5G1 MoAb and anti-CR3 NCI 5G11-2 as the capture antibody and Abcam rabbit anti-CR1 MoAb (ab108391) as the detector component. 

Reviewer 2 Report

Comments and Suggestions for Authors

Authors report the generation of Cripto-1/Cripto-3 specific monoclonal antibodies (mAbs) using the mice immunization/Hybridoma approach. Detection of Cripto-3 versus crypto-1 can have still unexplored diagnostic/therapeutic relevance in the field of cancer, as the protein Cripto-1 is raising since decades a strong interest as biomarker and therapeutic target.

 mAbs have been generated immunizing mice with short peptides differing for the few residues characterizing the primary structure deviation between the two proteins.

The mAbs specifically detect the two proteins at pg/mL levels in solution, in IHC tests and in WB analyses, suggesting their further use as valuable diagnostic tools (so far at lab scale) for detecting/measuring the proteins in liquid and solid biopsies.

The manuscript is interesting as it opens new avenues of investigations on this protein(s) family whose role in biology is still elusive despite the 3-decade long history of studies and incomplete therapeutic developments of inhibitors.

Minor changes are however required to accept the paper in a final form.

One major concern regards the Biacore binding assays, as the author claim sub-nanomolar affinity for the anti-Cripto-1 mAb which is apparently not reflected by the dissociation curves in the sensorgrams (Fig. 3A). Also the estimated KD for Cripto-3 is estimated as 0.6 nM outside the used solute concentrations between 0.8 -> 500 nM. Authors must explain or recalculate the KD.

In all work please use 50ng/50ul, not 50ng/50ml

Figure 2 Elisa:

Concentrations should be put instead of dilutions

Figure3 SPR:

NCI 5G1-1 antibody should be used up to 40nM. It will also improve the KD

Table2 missing SD on the KD

Table 3 put the ug/ml not the dilutions

Please provide a WB with their own or commercial recombinant proteins with respective antibodies.

In serum assay: is it significant? Shouldn't there be more crypto in tumors?

Author Response

Reviewer 2

Reviewer's Concern 1 - One major concern regards to the Biacore binding assays, as the author claims sub-nanomolar affinity for the anti-Cripto-1 mAb which is apparently not reflected by the dissociation curves in the sensorgrams (Fig 3A).  Also, the estimated KD for Cripto-3 is estimated as 0.6 nM outside the used solute concentration between 0.8 0 -> 500 nM.  Authors must explain or recalculate the KD. 

Author's Correction - We have changed the "E" notation of the Biacore program calculations in Table 2 into number x 10 power for better clarity. 

We do not claim our anti CR1 antibody (NCI 5G1-1) has subnanomolar affinity.  Its KD = 32 nM which is still a fairly good binding antibody with exquisite specificity to solid phase CR1 and showing no binding to CR3 even at 500 nM.

Our anti-CR3 antibody (NCI 511-2) was also tested from 0.8 nM to 500 nM.  It is my understanding that the Biacore program does what is equivalent to a "Linear Regression" analysis with the data from the collective antibody concentrations tested and can project a KD outside of the concentration range evaluated.  Because both the kon and koff for this antibody are with the instrument’s detection limits, the KD value 0.6 nM is still valid.  We are including the zoomed-in graph of NCI 511-2 for the reviewer's reference.  At the lowest concentration 0.8 nM, the antibody clearly has significant binding (black arrow) and shows very slow dissociation (blue arrow), a hallmark of a high affinity antibody.

Reviewer's Concern 2 -In all work please use 50ng/50ul, not 50ng/50ml.

Author's Correction - Appropriate corrections to 50ng/50ul have been made throughout the manuscript.

Reviewer's Concern 3 - Figure 2 ELISA:  Concentrations should be put instead of dilutions.

Author's Correction - In Figure 2 ELISA concentration have be incorporated according to Review 2's request.

Reviewer's Concern 4 - NCI 5G1-1 antibody should be used up to 40nM.  It will also improve the KD.

Author's Correction - In the Biacore analysis for both NCI 51-1 (anti-CR1) and NCI 511-2 (anti-CR3) MoAbs were run at 0.8, 4, 20, 100 & 500 nM, hence the Reviewer's requested 40 nM range was convers withing the progressive MoAb concentrations evaluated. 

Reviewer's Concern 5 - Table 2 missing SD on the KD.

Author's Correction - There are no SD cited in the Table 2.  This table  has been revised to eliminate any confusion the "E" notations of the original Biacore program calculation possible caused - see response to Reviewer's Concern 1 above. 

Reviewer's Concern 6 - Table 3 put the ug/ml not the dilution.

Author's Correction - Corresponding ng/ml antibody concentrations for individual antibody dilutions have been added to modified Table 3.

Reviewer's Concern 7 - Please provide a WB with their own or commercial recombinant proteins witih respective antibodies.

Author's Correction - We understand completely why the Reviewer has requested this modification to our WB analysis but it is much more complicated than simply rerunning the blot.  We are in the process of characterizing the 50 kDa CR1 product identified in our initial WB using use monensin/tunicamycin to block in vivo glycosylation or N-glycosidase to directly cleave sugar residues as was reported by Brandt et al. (reference 50 of the manuscript).  Interestingly when assessing what recombinant standards to run, it became evident that we could use the R&D Systems recombinant Baculovirus CR1 protein (insect cells) which potentially could N-glycosylate proteins but not identical to attached sugar residues that would occur in mammalian cells.  Similarly, MyBioSourse has recombinant proteins from E.coli, Yeast, Baculovius and Mammalian Cells.  In our follow up study on the 50 kDa CR1 product mention above we are planning to run all commercially available standards for a direct comparison.  Our initial Cancer manuscript is an introductory overview on the characterizing/validating of our selective MoAbs to CR1 vs CR3 using a variety of assay systems.  We feel adding recombinant standards to our WB will not alter our collective findings on specificity.

Reviewer's Concern 8 - In serum assay:  is it significant?  Shouldn't there be more cripto in tumors?

Author's Correction - We agree with the Reviewer's concerns regarding our observed levels of CR1 vs CR3 in serum samples from breast cancer.  In both cases the observed values are lower than for normal donor serum. Given that our IHC of paraffin embedded breast cancer tissue show relatively intense staining for both CR1 and CR3, one would expect higher levels for both proteins in the cancer donor serum also.  Unfortunately, this is not what we observed.  We mention in the discussion that our sample size was small - 10 normals and 10 breast CA.  In the discussion we also mention a possible caveat in the design of our initial capture/quantitative ELISA that could explain these results. 

Reviewer 3 Report

Comments and Suggestions for Authors

In this manuscript Cuttitta and colleagues report the development of mouse monoclonal antibodies that can differentiate human CRIPTO 1 and CRIPTO 3 proteins with high affinity binding, and no measurable cross-reactivity. Different characterization studies have been carried out that include solid phase ELISA, Biacore and IHC/WB analysis.  This is an interesting topic, the new Abs described are important tools to provide insights into the complexity of CRIPTO biology in future studies. There are some points that need to be addressed prior to publication.

The authors correctly mention that the low number of samples evaluated is a limitation of their study that should be taken into consideration; see for instance 3.6. Western Blot Analysis of Human Tumor Cell Lines. In this context, the authors need to provide replicates and statistical significance of the WB data (Fig. 6 panel C) to support their conclusion “It appears on average, that CR1 represents the highest protein expression when compared to CR3 in most of the cell lines examined, with the exception of HepG2.

3.4. Immunohistochemistry (IHC) Results. Some colorectal adenocarcinoma tumor cells stained for CR3 but not for CR1, while others stained positive for both. Is there any correlation with different genetic background of the tumors?

Line 724-725. “ the preponderance of published reports over the past thirty years get CR1 as the major oncofetal protein involved in embryogenesis, wound repair and carcinogenesis but our current data suggests otherwise.” This sentence should be revised/toned down. Indeed, the main role of CR1 in embryogenesis and tissue regeneration is revealed using KO strategies that specifically target CR1, and thus it is reasonable to conclude that it cannot be compensated by expression of CR3, if any. Future studies are needed to address the role of CR3 using targeting strategies and functional studies that are missing.

Line 743-746  “the expression of CR1 in the in vitro setting appears to be completely opposite of what is seen in vivo” This conclusion should be toned down due to the low number of samples analyzed in this study, see comment above.

Line 754-755. The closing statement should be revised. While the study presents intriguing preliminary data showing CRIPTO 3 expression in both tumors and normal tissue, it lacks the functional studies needed to substantiate the conclusion that CRIPTO 3 plays a significant role in CRIPTO biology.

Introduction, page 2. The authors should refine the references and includes the work showing the role of Cripto in regulating stem cells dynamic heterogeneity in physiological and pathological conditions ( tissue regeneration and different types of solid tumors), a process that is gaining increasing interest in the literature (for example, Fiorenzano, A., et al. (2016). Nat. Commun. 7, 1258; Francescangeli, F., et al. (2015) Cell Death Differ. 22, 1700–1713; Guardiola et al., 2024 Developmental Cell). The references cited (13-16) refer to cancer stem cells and mostly limited to breast cancer tumor.

Author Response

Reviewer 3

Reviewer's Concern 1 - The authors correctly mention that a low number of samples evaluated is a limitation of their study that should be taken into consideration: see for instance 3.6 Western Blot Analysis of Human Tumor Cell Lines.  In this context, the authors need to provide replicates and statistical significance of the WB data (Fig 6 panel C) to support the conclusion "it appears on average, that CR1 represents the highest protein expression when compared to CR3 in most of the cell lines examined, with the exception of HepG2."

Author's Correction - Authors agree with the reviewer’s objections and have altered their written "Result" statement to reflect this correction.  "Given the small sample size of human tumor cell lines examined by WB and even through CR1 appears to be more prevalently expressed than CR3, no definitive conclusions should be drawn until a more diverse and larger tumor cell line number is examined".

Reviewer's Concern 2 -Immunohistochemistry (IHC) Results.  Some colorectal adenocarcinomas tumor cells stained for CR3 but not for CR1 while others stained positive for both.  Is there any correlation with different genetic background of the tumors?

Author's Correction - Very interesting question.  Given that CR1 is expressed on Chromosome 3 and CR3 on the X-Chromosome, one could potentially assume that females (XX) may have twice as much CR3 expression as males (XY).  When analyzing the IHC colorectal adenoCA results we don't see a gender bias reflected in the CR3 expression observed - validating the established convention that in female cells only one X-chromosome is actually read. 

Reviewer's Concern 3 - Line 724-725.  "the preponderance of published reports over the past thirty years get CR1 as the major oncofetal protein involved in embryogenesis, wound repair and carcinogenesis but our current data suggests otherwise".  This sentence should be revised/tone down.  Indeed, the main role of CR1 in embryogenesis and tissue regeneration is revealed using KO strategies that specifically target CR1 and thus it is reasonable to conclude that it cannot be compensated by expression of CR3, if any.  Future studies are needed to address the role of CR3 using targeting strategies and functional studies that are missing.

Author's Correction - We agree with the Reviewer's concern and have altered our lead sentence in our "Conclusion" to reflect their "Revise/Tone Down" request. 

Reviewer's Concern 4 - Line 743 - 746, "the expression of CR1 in the in vitro setting appears to be completely opposite of what is what is seen in vivo".  This conclusion should be toned down due to the low number of samples analyzed in the study.  See comment above.

Author's Correction - Here again, we agree with the Reviewer's concerns completely and have removed that statement from our ending "Conclusion" paragraph.

Reviewer's Concern 5- Line 754 - 755.  The closing statement should be revised.  While the study presents intriguing preliminary data showing CRIPTO 3 expression in both tumor and normal tissue, it lacks the functional studies needed to substantiate the conclusion that CRIPTO 3 plays a significant role in CRIPTO biology.

Author's Correction - We have completely removed the Reviewer's objecting sentence in our closing statement of our "Conclusion". 

Reviewer's Concern 6 - page 2.  The authors should refine the reference and include the work showing the role of Cripto in regulating stem cells dynamic heterogeneity in physiological and pathological conditions (tissue regeneration and different types of solid tumors), a process that is gaining increasing interest in the literature (for

example, Fiorenzano, A., et al (2016). Nat. Commun. 7, 1258; Francescangeli, F., et al. (2015) Cell Death Differ. 22, 1700-1713; Guardiola, O., et al. 2024 Developmental Cell).  The references cited (13-16) refer to cancer stem cells and mostly limited to breast cancer tumor.

Author's Correction -Reviewer's cited references on Cripto involvement in stem cell dynamic heterogeneity have now been incorporated into our manuscript reference list.
